# FEATURE WARPING-AND-CONDITIONING FOR REPRESENTATION-GUIDED NOVEL VIEW SYNTHESIS

## ABSTRACT

We propose a novel framework for diffusion-based novel view synthesis, harnessing the rich semantic and geometric representations provided by VGGT, a transformer foundation model for geometry prediction. Unlike existing methods that either rely on explicit 3D models or monocular depth estimates, our approach reformulates view synthesis as a warping-and-inpainting task: first, features from reference views are geometrically warped into a target pose; then, a diffusion model generates the final image by attending to both warped features for accurate reconstruction of visible regions and semantically similar cues for plausible inpainting of occluded areas. Through an empirical analysis of various diffusion foundation models such as DINOv2, CroCo, and VGGT, we demonstrate that VGGT's multiscale attention consistently delivers superior geometric correspondence and semantic coherence. Building on these insights, we design a multi-view synthesis architecture, named **ReNoV**, achieving **Re**presentation-guided **No**vel **V**iew synthesis through dedicated warping-and-conditioning modules that inject VGGT features into the diffusion process. Our experiments show that this design yields marked improvements in both reconstruction fidelity and inpainting quality, outperforming prior diffusion-based novel-view methods on standard benchmarks and enabling robust synthesis from sparse, unposed image collections.

## 1 INTRODUCTION

Novel view synthesis—predicting scene appearance from target camera viewpoints—has long been a fundamental challenge in computer vision. Recent diffusion models enable novel view generation without explicit 3D representations such as Neural Radiance Fields (Mildenhall et al., 2021) or 3D Gaussian Splatting (Kerbl et al., 2023).

Recent advances in diffusion-based novel view synthesis have introduced warping-and-inpainting methods (Chung et al., 2023; Seo et al., 2024) as an emerging approach. These approaches typically employ off-the-shelf geometry estimation modules—such as monocular depth predictors or DUSt3R (Wang et al., 2024)—to first estimate camera poses and scene geometry from a reference image. The predicted geometric information guides spatial cross-attention within generation diffusion models by conditioning target viewpoint generation with warped reference coordinates (Seo et al., 2024), thereby enhancing geometric consistency between generated and the reference image.

This motivates a critical question: *what qualities characterize an optimal representation for warping-and-inpainting novel view synthesis?* We observe that inpainting from partially warped information divides novel view synthesis into two distinct tasks: faithful reconstruction of visible regions that can be warped from reference viewpoint, and plausible inpainting of regions occluded in the reference image. Through empirical analysis, we find that warping-and-inpainting diffusion models (Seo et al., 2024) exhibit spatial attention behaviors consistent with this insight: during reconstruction, the model seeks precise geometric correspondences with locations in the reference image, whereas during inpainting, it attends to semantically relevant features that guide generation to remain consistent with the reference view.

This distinction suggests that representation choice significantly impacts both reconstruction quality and inpainting coherence. We conduct in-depth analysis of major features (Oquab et al., 2023; Weinzaepfel et al., 2022; Wang et al., 2025) regarding their semantic and geometric feature awareness, as well as their novel view reconstruction capabilities from warped geometry. Our analysis reveals the

impressive capabilities of VGGT (Wang et al., 2025), a recent transformer-based geometry prediction model whose rich, multiview-consistent, semantic and geometric features make it suitable for novel view conditioning and generation from multiple reference images.

In this light, we introduce a novel architecture that leverages VGGT's powerful geometric and semantic features for novel view image prediction. We design a multi-view synthesis architecture where a reference network extracts features from multiple source views, which are then aggregated with the target-view generation features via attention in a generation network. To enhance reconstruction and inpainting performance at target viewpoint generation, we introduce a **feature warping-and-conditioning** paradigm, which geometrically warps VGGT-extracted reference features to the novel viewpoint, providing conditioning guidance to improve the diffusion model's synthesis quality, and propose **ReNoV** (**Re**presentation-guided **No**vel **V**iew synthesis). Extensive experiments on RealEstate10K and zero-shot evaluation on DTU demonstrate that our method shows competitive results to state-of-the-art feedforward novel view synthesis approaches across both interpolation and extrapolation settings, with ablation studies confirming the effectiveness of our integrated semantic and geometric conditioning approach.

## 2 RELATED WORK

**Diffusion-based 3D generation models.** Prior efforts in generative 3D and multi-view synthesis have largely focused on leveraging diffusion models to bridge the gap between 2D image priors and 3D scene representations. DreamFusion (Poole et al., 2022) first demonstrated text-to-3D generation by optimizing a Neural Radiance Field with a pretrained 2D diffusion prior, while ProlificDreamer (Wang et al., 2023) extended this paradigm by distilling multi-view diffusion signals into a feed-forward geometry network for faster inference. In the multi-view setting, MVDream (Shi et al., 2023) proposes a view-consistent denoising pipeline that jointly refines color and depth across posed images, and Zero123 (Liu et al., 2023) tackles single-image to novel-view synthesis via a conditioned diffusion model that hallucinates plausible viewpoints. More recently, Marigold (Ke et al., 2024) integrates implicit surface representations with diffusion-based image priors to yield high-fidelity 3D reconstructions from sparse views. While these methods have achieved impressive visual quality, they either require costly per-scene optimization, rely on known camera poses, or struggle with large pose extrapolation.

**Feedforward 3D regression models.** Feed-forward approaches to novel-view synthesis and 3D reconstruction bypass costly per-scene optimization by learning rich geometric priors from large-scale training. PixelNeRF (Yu et al., 2021) first demonstrated how to condition a NeRF on input views via local CNN features, and IBRNet (Wang et al., 2021) built on this by fusing multi-view depth and appearance cues in a self-supervised stereo framework. MVSplat (Chen et al., 2024) further refines this paradigm by estimating 3D Gaussians through cost-volume–based depth prediction, achieving high-fidelity volumetric representations from sparse inputs. More recently, transformer-based systems such as DUSt3R (Wang et al., 2024) and MASt3R (Leroy et al., 2024) have learned to predict point-maps and camera poses directly from unposed images, while Noposplat (Ye et al., 2024) unifies pose estimation with 3D Gaussian fitting in a single feed-forward pass. Concurrently, single-image methods like ShapeFormer (Yan et al., 2022) exploit transformer architectures to hallucinate novel views and coarse geometry from one shot. Despite their efficiency, these feed-forward models remain fundamentally limited by reference-view visibility, often failing to extrapolate to unseen angles or complete occluded structures without explicit inpainting or geometry completion.

## 3 MOTIVATION AND ANALYSIS

As discussed in Sec.2, novel view synthesis approaches fall into several categories. Non-generative approaches—e.g., MVSplat(Chen et al., 2024) and NopoSplat (Ye et al., 2024)—do not exploit generative models and therefore cannot infer geometry or appearance in regions unseen or occluded in the reference images. In contrast, diffusion-based generative methods can extrapolate to viewpoints distant from the inputs; however, as these methods condition the diffusion models on target camera pose as a feature embedding, they remain confined to the pose distribution encountered during training, precluding truly arbitrary novel-pose synthesis.

We interpret novel view synthesis as a warping-and-inpainting problem, akin to GenWarp (Seo et al., 2024), requiring models to excel at two tasks: accurate *reconstruction* of visible regions and consistent *inpainting* of occluded regions. Within diffusion-based frameworks, both reconstruction and inpainting are achieved by implicitly aggregating features from reference viewpoints through the U-Net's spatial attention modules, driven by conditioning features that establish cross-view correspondences. This naturally leads to the question: what properties should an ideal conditioning feature possess for effective novel view generation?

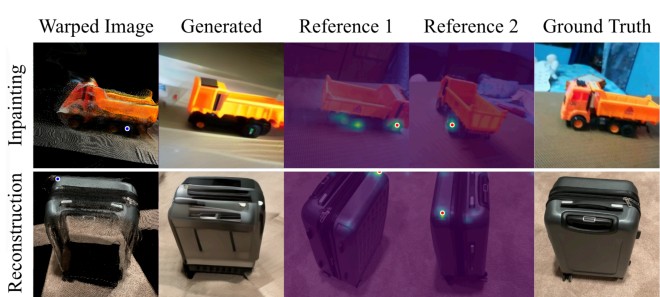

To this end, we examine diffusion-model attention during novel-view synthesis and uncover a consistent pattern (Fig. 1): regions visible in the reference views—those requiring reconstruction—attend sharply to their geometric correspondences, whereas regions needing inpainting attend broadly to semantically similar locations in the references. This can be intuitively understood, as reconstruction performance hinges on pinpointing exact correspondences, while inpainting relies on semantically related context to synthesize unseen areas coherently. This motivates the search for a conditioning representation that simultaneously encodes semantic awareness and geometric correspondence. In the next section, we evaluate several representations (Oquab et al., 2023; Weinzaepfel et al., 2022; He et al., 2022; Wang et al., 2025) to identify the representation that best balances semantic awareness with geometric correspondence, and offer a comprehensive analysis. To identify the optimal conditioning feature for our warping-and-inpainting diffusion framework, we compare several widely-used representations—DINOv2 (Oquab et al., 2023), CroCo (Weinzaepfel et al., 2022), and VGGT (Wang et al., 2025).

Figure 1: **Cross-view attention maps of the denoising network (Seo et al., 2024).** A query pixel (blue dot) is chosen in the warped target view, and the resulting cross-attention weights on two reference images are visualized. In the Inpainting: the wheel is absent in the warped view, so attention shifts to the corresponding wheels in the references. Reconstruction: the suitcase edge is visible, so attention concentrates on the geometrically aligned edges to refine the reconstruction.

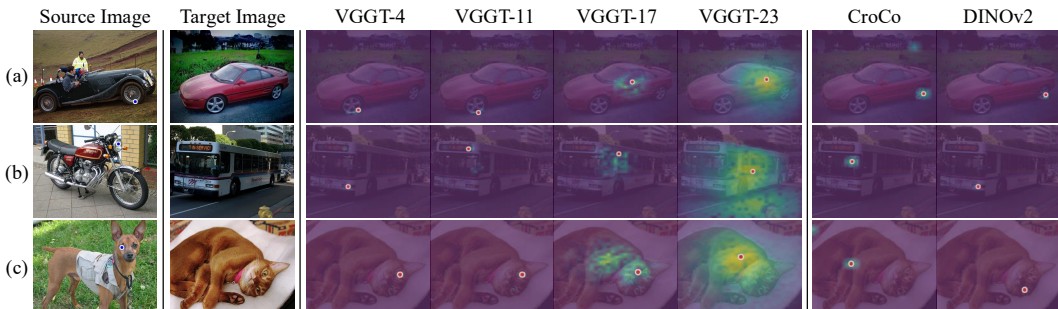

Figure 2: **Visualization of feature similarity map.** The leftmost column shows the source image with a query point (blue dot), followed by the target image. Cosine similarity is computed between the query and all target patch features to assess semantic encoding. Early VGGT layers (4, 11) retain strong semantic signals, effectively highlighting fine-grained regions (e.g., wheel, headlight, eye), while deeper layers (17, 23) lose semantic focus. DINOv2 captures rich semantics but with less precise localization. CroCo fails to capture meaningful cues, often highlighting irrelevant regions.

**Semantic consistency.** VGGT is a transformer-based model that infers 3D scene geometry from multiple reference views. Specifically, its architecture is composed of a series of attention layers, of which features from four of the layers (4th, 11th, 17th and 23th) are aggregated and put into a DPT (Ranftl et al., 2021) model for geometry prediction. We hypothesize that VGGT's ability to establish rich inter-view correspondences endows it with the semantic and geometric properties essential for effective novel-view conditioning. To evaluate the extent to which semantic information is captured within the learned representations of VGGT (Wang et al., 2025), we conduct an analysis

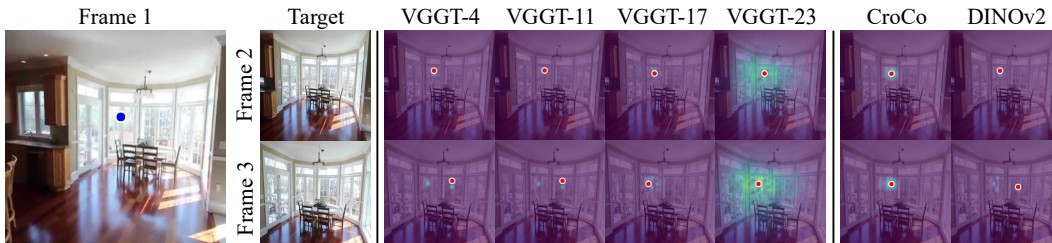

Figure 3: **Geometric correspondence evaluation.** A query point (blue dot) is selected in Frame 1, and cosine similarity maps are computed in Frame 2 and Frame 3. The scene contains repeated structures (e.g., identical windows), allowing assessment of whether the model can localize the geometrically corresponding instance. Deeper layers of VGGT (e.g., VGGT-23) and CroCo accurately identify the correct window aligned with the query point, while earlier layers (VGGT-4, VGGT-11) and DINOv2 attend to incorrect but semantically similar windows. This illustrates that deeper layers of VGGT, as well as CroCo, capture geometric structure more reliably than others.

of its ability to establish semantically meaningful correspondences across pairs of images. As VGGT (Wang et al., 2025) was not explicitly trained to predict inter-image correspondences, its ability to match semantically consistent pixels across diverse exemplars reveals the emergent semantic properties captured by its learned representations.

Specifically, given a source–target image pair $I_{src}$ and $I_{tgt}$ from different categories, we acquire feature tensors $f_{src} \in \mathbb{R}^{\frac{H}{P} \times \frac{W}{P} \times C}$ and $f_{tgt} \in \mathbb{R}^{\frac{H}{P} \times \frac{W}{P} \times C}$ through the VGGT model. We then locate a single query point in $f_{src}$, and compute its feature's cosine similarity between every feature vector in $f_{tgt}$, yielding a similarity map of size $H/P \times W/P$. We qualitatively evaluate the similarity maps across different layers of VGGT using the SPair-71k (Min et al., 2019) dataset. Our observations show that early layers in VGGT retain strong semantic signals. For example, as shown in Fig. 2a), when the query point is located on the *front wheel* of a car, VGGT layer 4 and 11 features successfully identify the corresponding *front wheel* in the target image, whereas layer 17 and 23 fail to do so. DINOv2 (Oquab et al., 2023) also encodes rich semantic representations; however, it lacks spatial awareness, often failing to distinguish directional context (e.g., incorrectly capturing the *back wheel* instead of the *front*). In contrast, CroCo (Weinzaepfel et al., 2022) fails to capture meaningful semantic information, frequently highlighting irrelevant or inconsistent regions. The initial layers of VGGT demonstrate a notable capacity to capture fine-grained semantic information.

**Geometric consistency.** We evaluate the extent to which each model captures geometric correspondence using a triplet of images from the same scene. A query point is selected in the first frame, and similarity maps are computed across the second and third frames. Each scene includes multiple instances of the same semantic object—for example in Fig 3, several identical windows arranged side by side—allowing us to evaluate whether the model can precisely localize the object instance that corresponds geometrically to the query point. We find that the deeper layers of VGGT (layers 11 and 23) effectively capture geometric structure, attending to the correct window that is spatially aligned with the query point in subsequent frames. In contrast, earlier layers often attend to an incorrect but visually similar window farther along the wall, suggesting they rely more on semantic similarity than spatial alignment. In this setting, CroCo (Weinzaepfel et al., 2022) demonstrates strong geometric consistency, accurately identifying the correct object, whereas DINOv2 (Oquab et al., 2023) frequently fails to disambiguate between repeated structures, revealing a lack of geometric awareness. VGGT processes multiple frames jointly and leverages its global attention mechanism to capture geometric structure consistently across views, enabling precise localization of the corresponding object instance even in the presence of repeated or ambiguous patterns.

**Feature reconstruction probing.** We have shown the fused VGGT feature can effectively encode semantic as well as geometric information. We assume that these properties play a pivotal role in conditioning warping-

| Model | PSNR ↑ | | | SSIM ↑ | | |
|---|---|---|---|---|---|---|
| | 1 view | 2 view | 3 view | 1 view | 2 view | 3 view |
| CroCo Weinzaepfel et al. (2022) | 15.28 | 15.37 | 15.47 | 0.440 | 0.434 | 0.435 |
| DINOv2 Oquab et al. (2023) | 14.95 | 15.35 | 15.37 | 0.537 | 0.532 | 0.527 |
| VGGT Wang et al. (2025) | **15.81** | **16.01** | **16.13** | **0.552** | **0.540** | **0.534** |

Table 1: **Quantitative evaluation for feature analysis.** We evaluate reconstruction NVS capability of each feature.

and-inpainting approach. To verify the hypothesis, we train a shallow MAE (He et al., 2022) decoder to predict a target view image from warped feature of reference view image. The optimal feature representation should encapsulate multi-view semantic and geometric information, enabling the model to accurately reconstruct visible regions while effectively inpainting occluded areas.

For DINOv2 (Oquab et al., 2023) and CroCo (Weinzaepfel et al., 2022), we probe the encoder output, whereas for VGGT we extract features from the 4th, 11th, 17th, and 23rd transformer layers and use all of them in our analysis. To facilitate feature warping, we employ an off-the-shelf geometry prediction model (Wang et al., 2025) to obtain the pointmaps and camera poses. The token-level features are re-projected into the target view; patches without valid projections are replaced by learnable mask tokens, and training is supervised with a mean-squared-error objective.

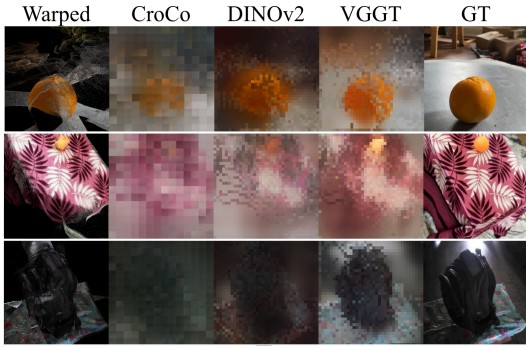

Figure 4: **Qualitative results for feature analysis.** We warp the extracted features using a predicted point cloud, resulting in feature-level holes that requiring inpainting at the feature level. VGGT feature synthesizes target view images with most accurate structure and color.

For quantitative results, we evaluate each model for different number of reference view using PSNR and SSIM metric. Table 1 show that VGGT feature consistently achieves the highest results across all metrics and inference settings. As the number of reference view is increase, the reconstruction capability is also improved. In the qualitative result, Fig. 4 demonstrates that the generated images using VGGT feature are most visually accurate with the target view images. Notably, VGGT outperforms other models in terms of preserving geometric structures and achieving realistic pixel colors, indicating its superior effectiveness for warping-and-inpainting approach.

## 4 METHOD

### 4.1 OVERVIEW

Our objective is to predict novel view image $I_{\text{tgt}}$ for target viewpoint $\pi_{\text{tgt}}$, combining the diffusion model's generative capabilities for consistent and realistic novel-view prediction and VGGT feature's powerful semantic and geometric correspondence capabilities. We assume $N$ unposed and sparse reference images are given, so that $\mathcal{I}_{\text{ref}} = \{I_n \in \mathbb{R}^{H \times W \times 3}\}_{n=1}^{N}$. Our framework, following (Seo et al., 2024), is composed a reference U–Net and a denoising U–Net, an architecture that resembles ControlNet (Zhang et al., 2023). Likewise, the generative denoising network is conditioned by features extracted from reference network, which in this corresponds to features from reference images.

The reference U–Net processes the input images, geometry and VGGT feature to extract multiview features, while the denoising U–Net generates the target image by refining a noisy latent, guided by warped features and geometric priors from the reference network.

### 4.2 REFERENCE CONDITIONING

**Geometry conditioning.** We begin by leveraging an off-the-shelf geometry prediction model (Wang et al., 2025) to estimate a set of camera poses $\{\pi_n \in \mathbb{R}^{4 \times 4}\}_{n=1}^{N}$ and corresponding pointmaps $\{P_n \in \mathbb{R}^{H \times W \times 3}\}_{n=1}^{N}$, where each $P_n$ is a 2D grid of 3D points representing the predicted world coordinates for the pixels of the reference image $I_n$. To incorporate geometric priors into our model, we apply a positional embedding function $\gamma(\cdot)$ to each pointmap, resulting in Fourier-encoded features $\gamma(P_n)$.

**VGGT feature conditioning.** We begin by jointly processing the $N$ reference images through the VGGT network, extracting features from the 4[th], 11[th], 17[th], and 23[rd] transformer layers. For each reference image $I_n$, we obtain both local and global features at each selected layer, denoted as $t_{l,n}$ and $t_{g,n} \in \mathbb{R}^{H/P \times W/P \times 1024}$, respectively. These features are concatenated along the channel

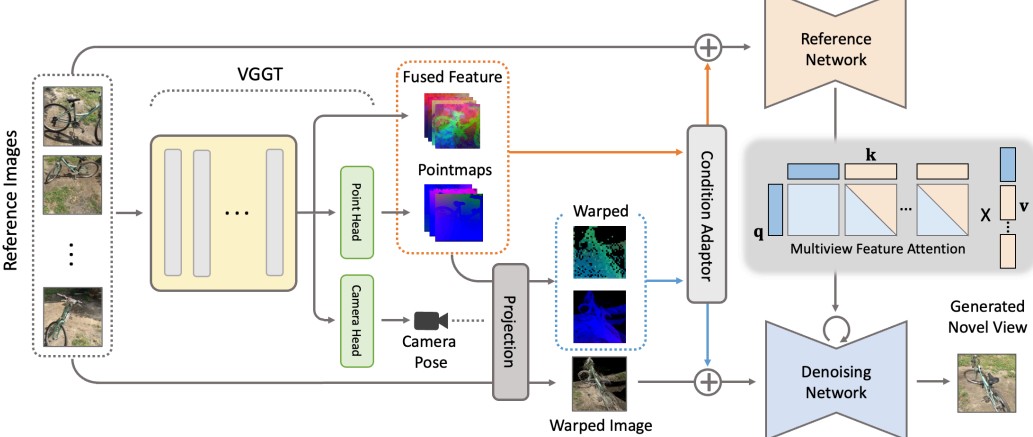

Figure 5: **Model architecture. Reference network (upper path)**: $N$ reference images are passed through VGGT, producing layer–wise visual features, per–pixel point-maps, and camera pose estimates. These outputs are fused by a lightweight reference U–Net to form a multiview reference feature. *Projection*: the features and point-maps are re-projected to the target camera frustum, yielding a warped RGB image and aligned feature planes. **Denoising network (lower path)**: a denoising U–Net receives a noisy latent together with the warped image and projected feature planes; at each timestep it mixes the feature from reference network with its own generation features through cross-attention, progressively refining the latent to synthesize the novel target view.

dimension to form a unified representation $T_n = [t_{g,n}; t_{l,n}] \in \mathbb{R}^{H/P \times W/P \times 2048}$. Yet, the obtained feature is high-dimensional, exceeding what the reference U–Net can efficiently process. To address this, $T_n$ is passed through a convolutional network.

Analogous to the pointmap conditioning, we apply a positional embedding function $\gamma(\cdot)$ to the extracted features $T_n$, resulting in the Fourier-encoded representation $\gamma(T_n)$. The final reference condition $c_n$ is obtained by concatenating the encoded image features and pointmaps:

$$c_n = [\gamma(P_n); \gamma(T_n)].$$

Following the approach of Hu et al. (Hu, 2024), each condition vector $c_n$ is passed through a shallow convolutional network and then added to the image latents prior to input to the reference U–Net.

### 4.3 GEOMETRY-AWARE FEATURE PROJECTION AND CONDITIONING

To enhance the fidelity of reconstruction and inpainting in novel view synthesis, we incorporate a geometry-driven conditioning mechanism based on warping. Specifically, we project the reference pointmaps $\{P_1, \ldots, P_N\}$ and the corresponding VGGT-derived features $\{T_1, \ldots, T_N\}$ into the target viewpoint $\pi_{\text{tgt}}$. These projected signals provide spatial priors that guide the diffusion model toward higher-quality generation results. First, the set of reference pointmaps $\{P_1, \ldots, P_N\}$, expressed in a global coordinate frame, can be directly aggregated to form a unified point cloud:$\mathcal{P}_{\text{ref}}$. This point cloud $\mathcal{P}_{\text{ref}} \in \mathbb{R}^{(N \times H \times W) \times 3}$ is then projected onto the target viewpoint $\pi_{\text{tgt}}$:

$$\mathcal{P}_{\text{tgt}}^{\Pi} = \Pi(\mathcal{P}_{\text{ref}}, \pi_{\text{tgt}}). \tag{1}$$

When multiple points are projected to the same pixel, only the one closest to the target image plane is retained, following the standard point cloud rasterization procedure (Seo et al., 2024). The resulting projected pointmap $\mathcal{P}_{\text{tgt}}^{\Pi}$ serves as a sparse geometric condition that guides the generation of $I_{\text{tgt}}$ from the reference views.

Given the multiview-consistent nature of VGGT features, we unproject them into 3D space by anchoring each pixel-level feature to its corresponding 3D coordinate from the predicted pointmap $P_n \in \mathbb{R}^{H \times W \times 3}$, forming a 3D feature point cloud. This pointcloud is then projected into the target view, yielding a spatially aligned warped feature map. The projected features $T_{\text{tgt}}^{\Pi}$ and projected pointmap $X_{\text{tgt}}^{\Pi}$ are provided as input conditions to the denoising network.

Following the same design as in the reference network, we encode $X_{\text{tgt}}^{\Pi}$ and $T_{\text{tgt}}^{\Pi}$ using a positional embedding function $\gamma(\cdot)$, and concatenate their Fourier embeddings with a binary visibility mask $M_{\text{tgt}}$, which indicates grid pixels where no 3D point was projected. This forms the target correspondence condition $c_{\text{tgt}}^d$:

$$c_{\text{tgt}}^d = [\gamma(X_{\text{tgt}}^{\Pi}), \gamma(T_{\text{tgt}}^{\Pi}), M_{\text{tgt}}].$$

The condition $c_{\text{tgt}}^d$ is then processed by a shallow convolutional network and added to the noise latent before being passed into the denoising U–Net. As discussed in Sec. 3, providing the warped feature $T_{\text{tgt}}^{\Pi}$ to the denoising U–Net serves two key purposes: it supplies semantic priors for unseen or occluded regions by leveraging multiview-consistent features, and it delivers accurate geometric information for regions visible in the reference views. This guidance enables the model to generate more structurally faithful outputs at the target view $\pi_{\text{tgt}}$.

### 4.4 NOVEL-VIEW IMAGE GENERATION

Following this, we conduct integrated self-and-cross attention between reference and target features, allowing the model to leverage other viewpoints, similar to (Seo et al., 2024). Specifically, from the denoising U-Net, we extract key and value features of the target view, $F_{\text{tgt}}^k, F_{\text{tgt}}^v \in \mathbb{R}^{1 \times C \times (W \times H)}$, obtained from spatial self-attention layers. from spatial self-attention layers. These are concatenated along the viewpoint dimension with key and value features from $N$ reference views, so that the query feature $\mathbf{q} = F_{\text{tgt}}^q$, is aggregated over attention map acquired with expanded key feature $\mathbf{k} = [F_{\text{tgt}}^k, F_1^k, \ldots, F_N^k]$ and value feature $\mathbf{v} = [F_{\text{tgt}}^k, F_1^v, \ldots, F_N^k]$. where $\mathbf{k}, \mathbf{v} \in \mathbb{R}^{(N+1) \times C \times (W \times H)}$. The aggregated attention is then computed as:

$$\text{Attention}(\mathbf{q}, \mathbf{k}, \mathbf{v}) = \text{softmax}\left(\frac{\mathbf{q}\mathbf{k}^T}{\sqrt{d_k}}\right)\mathbf{v}, \tag{6}$$

where $d_k$ denotes the dimensionality of the key features. Through this architecture, the generating U-Net can leverage features extracted from reference networks via attention aggregation, enabling novel view synthesis from multiple viewpoints.

## 5 EXPERIMENTS

### 5.1 IMPLEMENTATION DETAILS

For our image synthesis pipeline, we initialize from the pre-trained Stable Diffusion 2.1 model (Rombach et al., 2022). The reference feature extraction networks share identical architecture with the denoising U-Net but exclude timestep embeddings, as they are designed solely for semantic feature extraction rather than denoising operations.

| Method | Far-view Setting | | | Near-view Setting | | |
|---|---|---|---|---|---|---|
| | PSNR↑ | SSIM↑ | LPIPS↓ | PSNR↑ | SSIM↑ | LPIPS↓ |
| PixelSplat[†] (Charatan et al., 2024) | 13.03 | 0.486 | 0.414 | 11.57 | 0.330 | 0.634 |
| MVSplat[†] (Chen et al., 2024) | 12.22 | 0.416 | 0.423 | 13.94 | 0.473 | 0.385 |
| NopoSplat (Ye et al., 2024) | 13.58 | 0.393 | 0.545 | 14.04 | 0.414 | 0.503 |
| **ReNoV (Ours)** | **15.45** | **0.584** | **0.297** | **15.38** | **0.599** | **0.274** |

Table 2: **Zero-shot evaluation on the DTU (Jensen et al., 2014).** [†] denotes methods that require camera poses of the reference images.

Training is conducted on three multi-view datasets: RealEstate10K (Zhou et al., 2018) for diverse indoor/outdoor scenes, Co3D (Reizenstein et al., 2021a) for object-centric captures, and MVImgNet (Yu et al., 2023) for extensive multi-view imagery. We generate pseudo ground-truth geometry using VGGT (Wang et al., 2025), which provides both depth maps and normal predictions to establish reliable geometric supervision. During training, reference pointmaps enable three key operations: spatial warping of reference RGB images to target coordinates, explicit warping of VGGT-derived high-dimensional features between viewpoints, and establishment of geometric conditioning signals that guide generation. At inference, VGGT serves as our geometry estimation module, producing camera poses and pointmaps for target viewpoint projection. Additionally, VGGT provides rich semantic and geometric features that undergo geometry-aware warping, ensuring proper transfer of spatial and semantic information across viewpoints while maintaining geometric consistency throughout synthesis.

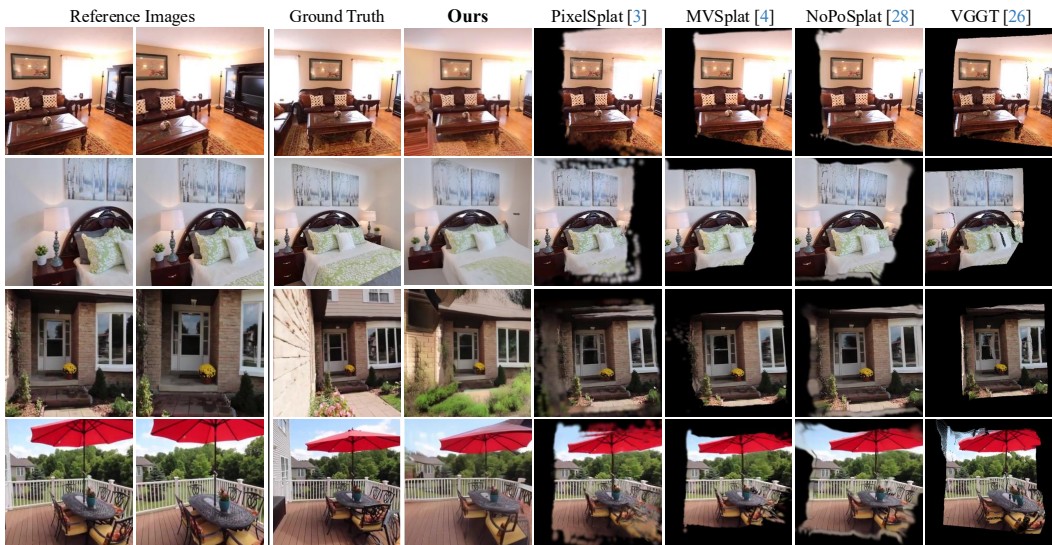

| Reference Images | Ground Truth | **Ours** | PixelSplat [3] | MVSplat [4] | NoPoSplat [28] | VGGT [26] |

Figure 6: **Qualitative comparison on extrapolative setting.** Qualitative results demonstrate our model's extrapolative capabilities to plausibly generate locations not seen in reference images while faithfully reconstructing the known regions.

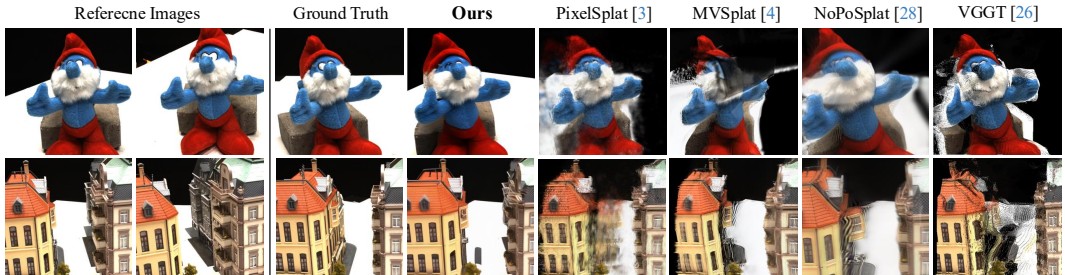

| Referecne Images | Ground Truth | **Ours** | PixelSplat [3] | MVSplat [4] | NoPoSplat [28] | VGGT [26] |

Figure 7: **Qualitative comparison on extrapolative setting.** Qualitative results demonstrate our model's extrapolative capabilities to plausibly generate locations not seen in reference images while faithfully reconstructing the known regions.

## 5.2 EXPERIMENT RESULTS

**Comparison with non-generative novel view synthesis models.** We compare our method with non-generative novel view synthesis models (Charatan et al., 2024; Chen et al., 2024; Ye et al., 2024) on RealEstate10K (Zhou et al., 2018) using a challenging far-view setting that requires extensive inpainting of missing regions. We evaluate on three target views conditioned on two reference views, with target cameras positioned far from reference cameras to create large unknown areas. As shown in Table 3, our method outperforms state-of-the-art approaches even without camera pose access. Non-generative methods struggle in this extrapolative setting due to their inability to generate unseen regions, being limited to fusing existing input views. In contrast, our diffusion-based approach enables strong performance on both interpolation and extrapolation tasks. The qualitative results (Fig. 6) demonstrate semantically plausible inpainting and accurate geometry reconstruction, attributed to features that incorporate both geometric and semantic information.

**Zero-shot evaluation.** We evaluate the generalization capability of our method using the DTU (Jensen et al., 2014) dataset, which was not seen during training. To comprehensively assess the generalization performance, we conduct evaluations under both near-view and far-view settings. For near-view, we follow the setting from MVSplat (Chen et al., 2024), while the far-view setting is constructed by selecting the farthest view as the target. Table 2 shows that our method outperforms previous methods (Charatan et al., 2024; Chen et al., 2024; Ye et al., 2024) across both settings. The qualitative results from Fig. 7 show that our method produces accurate geometry and semantically consistent inpainting, even in challenging target viewpoint of the out-of-domain data.

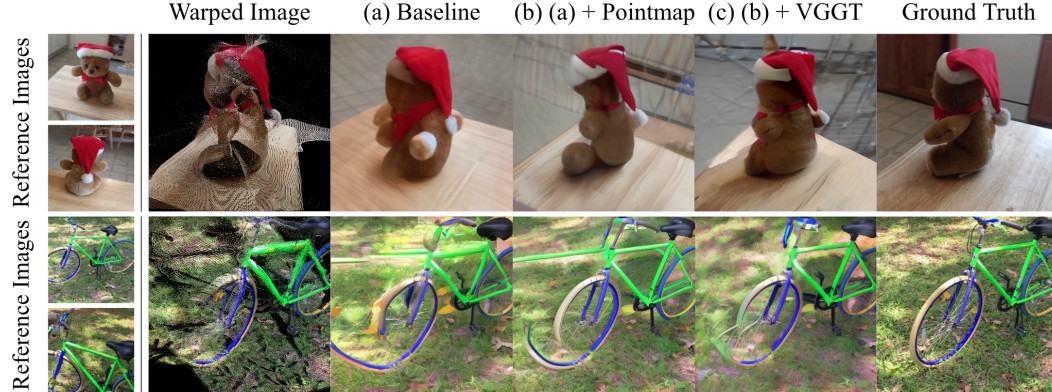

Figure 8: **Qualitative results for ablation study. Top row (bear)**: Both (a) and (b) fail to reconstruct structurally consistent outputs, exhibiting misaligned body parts such as the arms, legs, and hat. In contrast, (c) effectively preserves both semantic consistency and structural integrity, producing coherent reconstructions aligned with the ground truth. **Bottom row (bicycle)**: Both (a) and (b) exhibit noticeable distortions in the wheel structure and fail to inpaint occluded background. Meanwhile, (c) achieves more accurate structural reconstruction and background inpainting, demonstrating superior semantic and geometric consistency.

## 5.3 ABLATION

We explore how semantic and geometric conditioning features affect the performance of novel view synthesis. Specifically, we evaluate three configurations: (a) Baseline, utilizing semantic information from reference views via aggregated attention only; (b) Baseline with explicit geometric guidance using predicted pointmaps; and (c) our final model conditioned on implicit semantic and geometric information by VGGT features. Quantitatively, Table 4 shows that explicit geometry conditioning through pointmaps in (b) improves overall performance compared to the baseline. Furthermore, conditioning VGGT features in (c) results in significant performance gains, highlighting the effectiveness of implicit geometric and semantic conditioning for extrapolative synthesis.

| Method | PSNR↑ | SSIM↑ | LPIPS↓ |
|---|---|---|---|
| PixelSplat[†] (Charatan et al., 2024) | 14.01 | 0.582 | 0.384 |
| MVSplat[†] (Chen et al., 2024) | 12.13 | 0.534 | 0.380 |
| NopoSplat (Ye et al., 2024) | 14.36 | 0.538 | 0.389 |
| **ReNoV (Ours)** | **17.49** | **0.598** | **0.265** |

Table 3: **Experimental result a far-view setting.** [†] denotes methods that require camera poses of the reference images.

In the qualitative evaluation (Fig. 8), the baseline model (a) exhibits clear limitations in synthesizing structurally coherent novel views, resulting in perceptually distorted shapes and inconsistent reconstructions. Although explicit pointmap conditioning in (b) reduces geometric distortions, it still suffer from inaccurate inpainting due to insufficient semantic guidance. In contrast, our final configuration (c) utilizes VGGT features, which implicitly encode both semantic and geometric correspondences. This integrated conditioning allows the model to learn semantically consistent inpainting in challenging occluded regions, as well as structurally aligned reconstruction.

| Components | PSNR↑ | SSIM↑ | LPIPS↓ |
|---|---|---|---|
| (a) Baseline | 16.551 | 0.559 | 0.260 |
| (b) (a) + Pointmap condition | 16.930 | 0.594 | **0.243** |
| (c) (b) + VGGT Feature | **17.497** | **0.598** | 0.247 |

Table 4: **Quantitative results for ablation study.** Evaluation results shows that leveraging the pointmaps and VGGT features enhances novel view synthesis performance.

## 6 CONCLUSION

We introduce a diffusion-based novel-view synthesis framework that leverages VGGT's multi-view geometry features to unify precise reconstruction and semantically coherent inpainting. By reformulating synthesis as a warping-and-inpainting task and injecting VGGT features into a conditioned diffusion U-Net, our method achieves state-of-the-art fidelity on both visible and occluded regions, outperforming existing diffusion-based approaches across standard benchmarks. These results underscore the value of rich geometric priors in guiding generative models, and open avenues for future extensions toward dynamic scenes and real-time applications.

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

# APPENDIX

In Sec. B, we provide additional implementation details for our proposed method. In Sec. C, we present the results of additional analysis experiments to validate our approach. In Sec. D, we provide additional comparison to other baselines as well as additional qualitative ablation results and analysis.

## A  THE USE OF LARGE LANGUAGE MODELS

Large language models (LLMs) in this study served solely as tools for linguistic enhancement and editorial refinement of the written manuscript. LLM assistance was limited to sub-sentence and sentence-level modifications, including correction of grammatical mistakes and restructuring of specific phrases to improve brevity and scholarly presentation of content originally authored by the research team. The conceptual framework, research methodology, experimental procedures, data analysis, and scientific interpretations were conceived and executed entirely by the authors without computational language model input. LLMs played no role in generating research concepts, developing methodological approaches, or analyzing experimental outcomes. The authors assume complete accountability for all manuscript content, including portions that received LLM-based linguistic assistance.

## B  ADDITIONAL DETAILS

### B.1  TRAINING DETAILS

In our training procedure, we initialize the image denoising U-Net from the Stable Diffusion 2.1 model and fine-tune it on a combination of large-scale datasets including RealEstate10K Zhou et al. (2018), Co3D Reizenstein et al. (2021b), and MVImgNet Yu et al. (2023). The reference networks, which are architecturally identical to the image denoising U-Net (albeit without timestep embeddings), share the same initial weights and are trained solely to extract high-level semantic features from the input images. Ground-truth geometry is generated using an off-the-shelf geometry predictor, and only pointmaps from selected reference views are used during training for warping and proximity-based mesh conditioning. This strategy ensures that our model learns to synthesize both image and geometric representations in a mutually reinforcing manner.

To further stabilize training, we perform cross-modal attention instillation in a one-on-one fashion before combining the networks for joint training. This separate instillation phase allows the image and geometry branches to initially learn robust representations independently. Later, during simultaneous training, the geometry networks benefit from the deterministic cues provided by the image denoising network, which significantly improves consistency in geometry prediction. Our training schedule includes careful hyperparameter tuning, data augmentation, and regularization to mitigate overfitting while ensuring that the network generalizes well to unseen viewpoints.

## C  ADDITIONAL ANALYSIS

### C.1  SEMANTIC CORRESPONDENCE

**Qualitative.**  Additional qualitative results are presented in Figure 9, further illustrating that the early layers of VGGT Wang et al. (2025) encode rich semantic information, which gradually diminishes in deeper layers. These early layers also exhibit an ability to capture geometrically consistent semantics. For instance, in Figure 9(g), when the query point is placed on the left horn of a cow, VGGT accurately identifies the corresponding left horn in the target image. In contrast, DINOv2 Oquab et al. (2023) matches the right horn, ignoring spatial alignment, while CroCo Weinzaepfel et al. (2022) fails to capture meaningful semantic correspondence and highlights irrelevant regions. Similar patterns appear throughout Figure 9(c), (f) and (j), where early VGGT layers demonstrate direction-aware and spatially accurate semantic matching, often outperforming DINOv2 in both precision and structure-awareness.

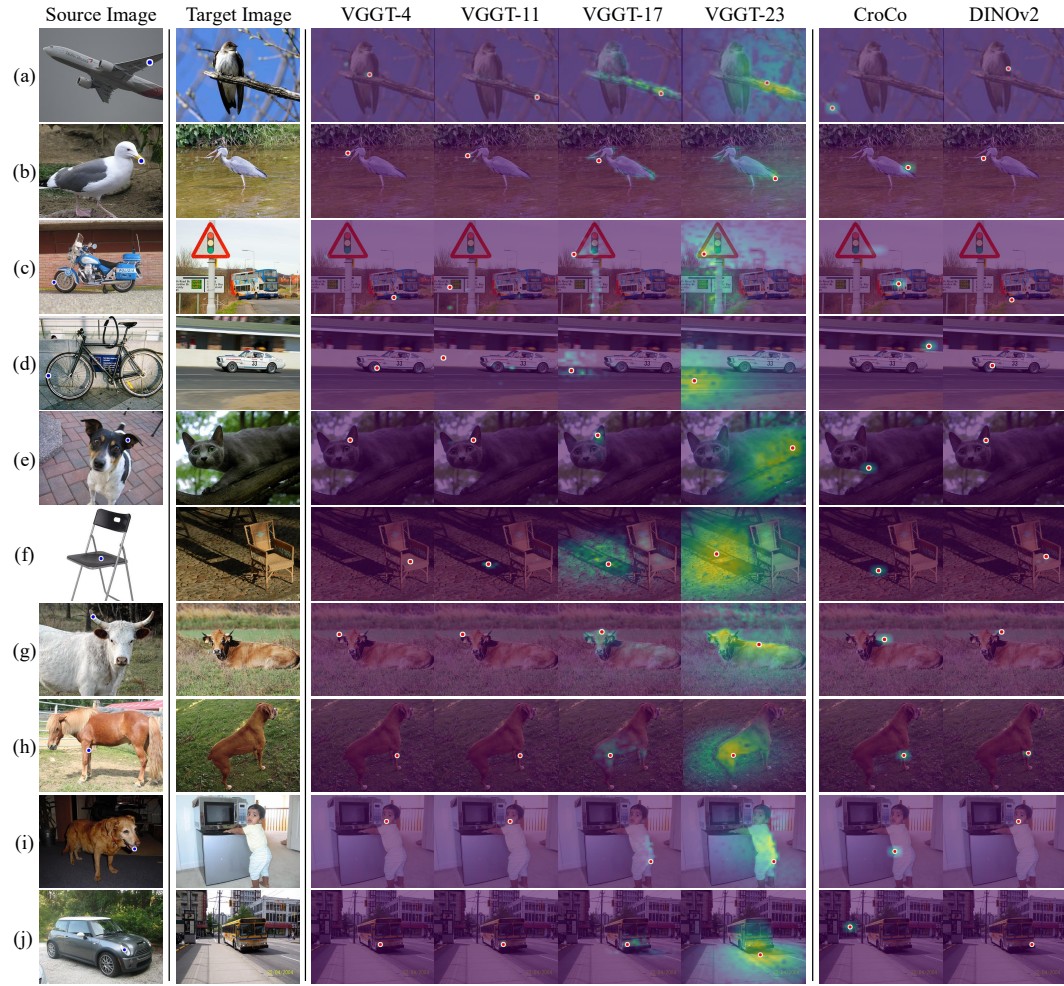

Figure 9: **Visualization of feature similarity map.** The leftmost column shows the source image with a query point (blue dot), followed by the target image. Cosine similarity is computed between the query and all target patch features to assess semantic encoding. Early VGGT layers ($4^{th}$, $11^{th}$) retain strong semantic signals, effectively highlighting fine-grained regions (e.g., wing, beak, wheel, ear, etc.), while deeper layers ($17^{th}$, $23^{rd}$) lose semantic focus. DINOv2 captures rich semantics but with less precise localization. CroCo fails to capture meaningful cues, often highlighting irrelevant regions.

## C.2 GEOMETRIC CORRESPONDENCE

Additional qualitative results are presented in Figure 10, highlighting that the deeper layers of VGGT Wang et al. (2025) better capture geometric structure, whereas the early layers often fail to establish accurate geometric correspondences. In scenes containing multiple instances of the same object—such as Figure 10(c), where several identical windows appear—features from layer 23 correctly match the query point above the first window across both frame 2 and frame 3. In contrast, earlier layers tend to match the query point to arbitrary windows, indicating a lack of geometric specificity. Similar trends are observed in Figures 10(a) through (g), where deeper layers consistently attend to the correct geometric location, while early layers often respond to semantically similar but spatially incorrect regions. Among baselines, CroCo Weinzaepfel et al. (2022) demonstrates strong geometric consistency, whereas DINOv2 Oquab et al. (2023) struggles to disambiguate repeated structures within the scene.

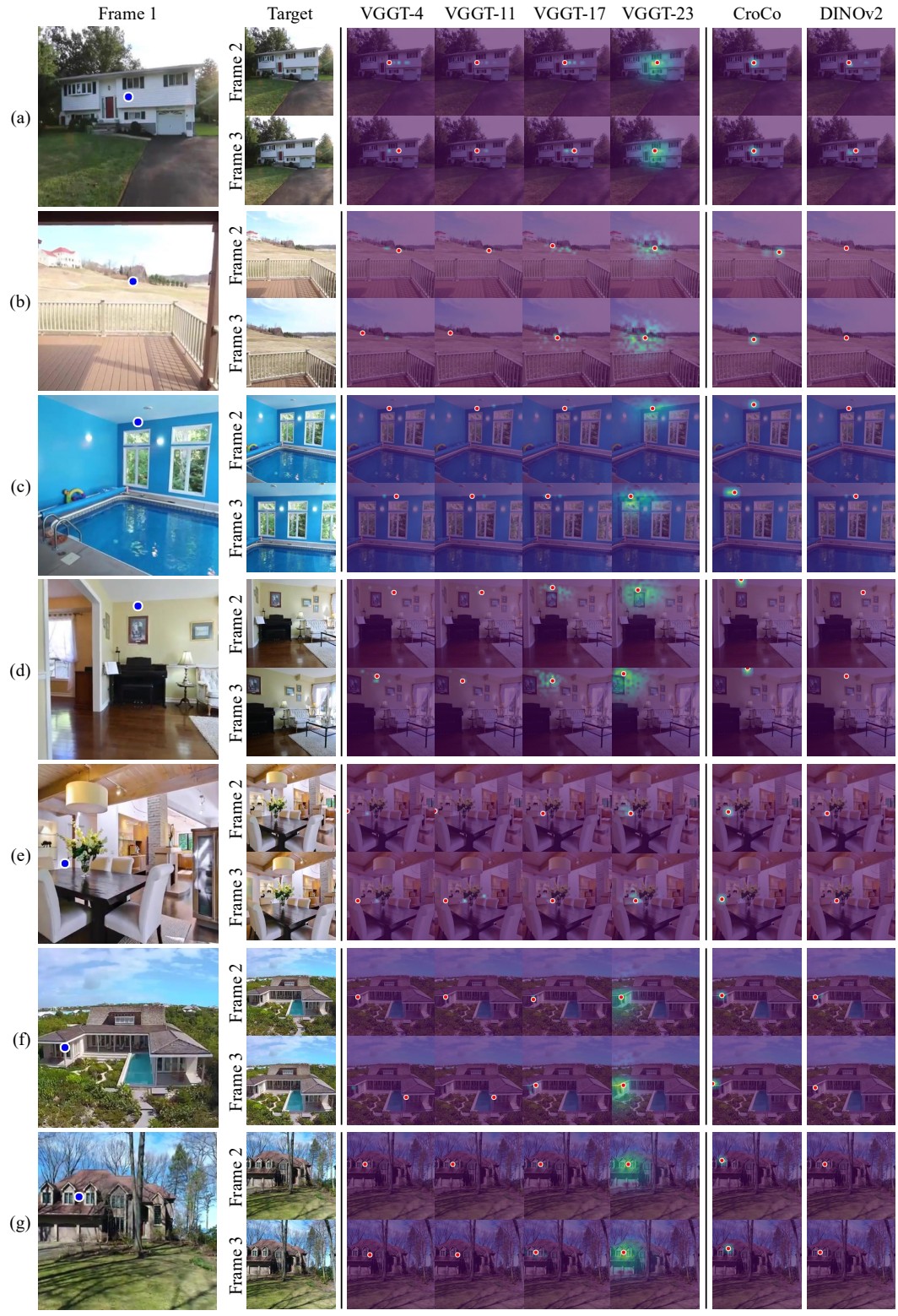

Figure 10: **Geometric correspondence evaluation.** A query point (blue dot) is selected in Frame 1, and cosine similarity maps are computed in Frame 2 and Frame 3. Each scene contains repeated structures (e.g., identical windows, mountains, frames, and columns), enabling evaluation of geometric alignment. Deeper layers of VGGT (e.g., VGGT-23) and CroCo accurately identify the correct window aligned with the query point, while earlier layers (VGGT-4, VGGT-11) and DINOv2 attend to incorrect but semantically similar position. This illustrates that deeper layers of VGGT, as well as CroCo, capture geometric structure more reliably than others.

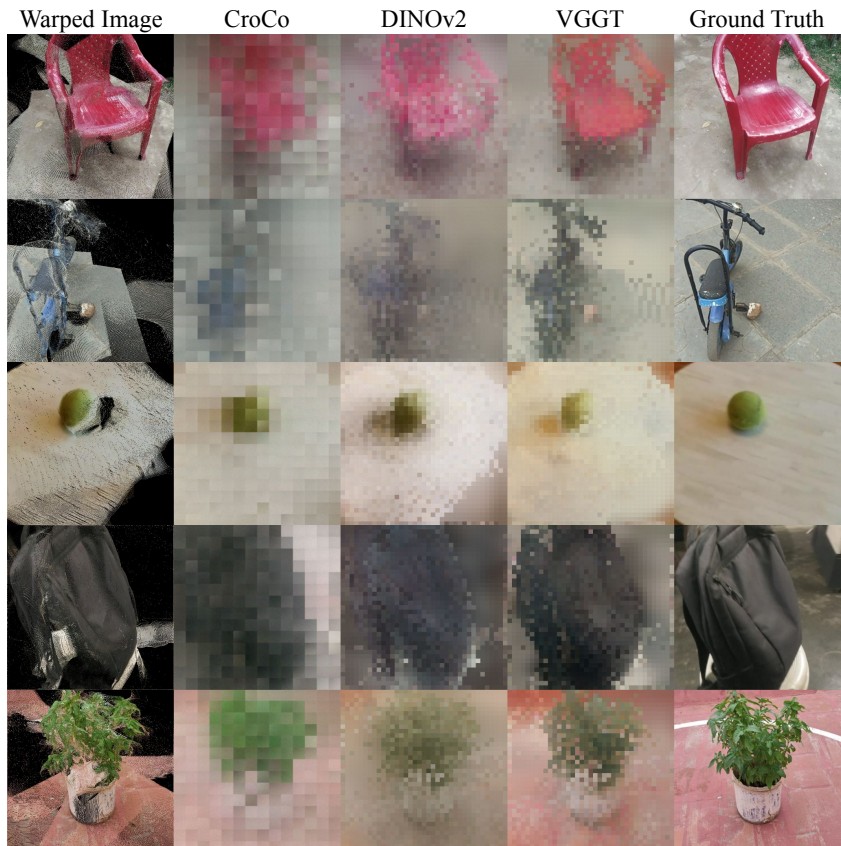

| Warped Image | CroCo | DINOv2 | VGGT | Ground Truth |

Figure 11: **Proving analysis qualitative results.** The leftmost column shows warped images, where features are warped using the same predicted pointmaps, resulting in corresponding feature-level holes. VGGT features yield the most accurate reconstruction in visible regions and demonstrate superior inpainting quality within occluded areas.

## C.3 FEATURE RECONSTRUCTION PROBING

To provide additional experimental validation of this hypothesis through systematic probing, we provide additional experimental results on probing, using a shallow MAE He et al. (2022) decoder trained to predict target view images from warped reference view features, as given in the main paper. The additional experimental results from this probing analysis offer further empirical evidence supporting our feature representation choices and their effectiveness in the warping-and-inpainting framework.

**Qualitative results.** Fig. 11 shows additional qualitative results for different features-Croco Weinza-epfel et al. (2022), DINOv2 Oquab et al. (2023), and VGGT Wang et al. (2025). Among them, images generated using VGGT features exhibit the highest geometric and semantic fidelity to the ground truth, highlighting VGGT's ability to effectively encode both multi-view geometric correspondences and rich semantic context.

**Ablation.** We conduct an ablation study to investigate the representational capability of VGGT Wang et al. (2025) features extracted from different layers. Specifically, we train a shallow MAE He et al. (2022) decoder on features from the $4^{th}$, $11^{th}$, $17^{th}$, and $23^{rd}$ layers, and evaluate their generation performance qualitatively. Fig. 12 demonstrates that deeper layers tend to capture more geometric structure but offer less semantic detail. In contrast, aggregating features across all layers results in the most visually plausible image, indicating effective reconstruction fidelity and more semantically coherent inpainting.

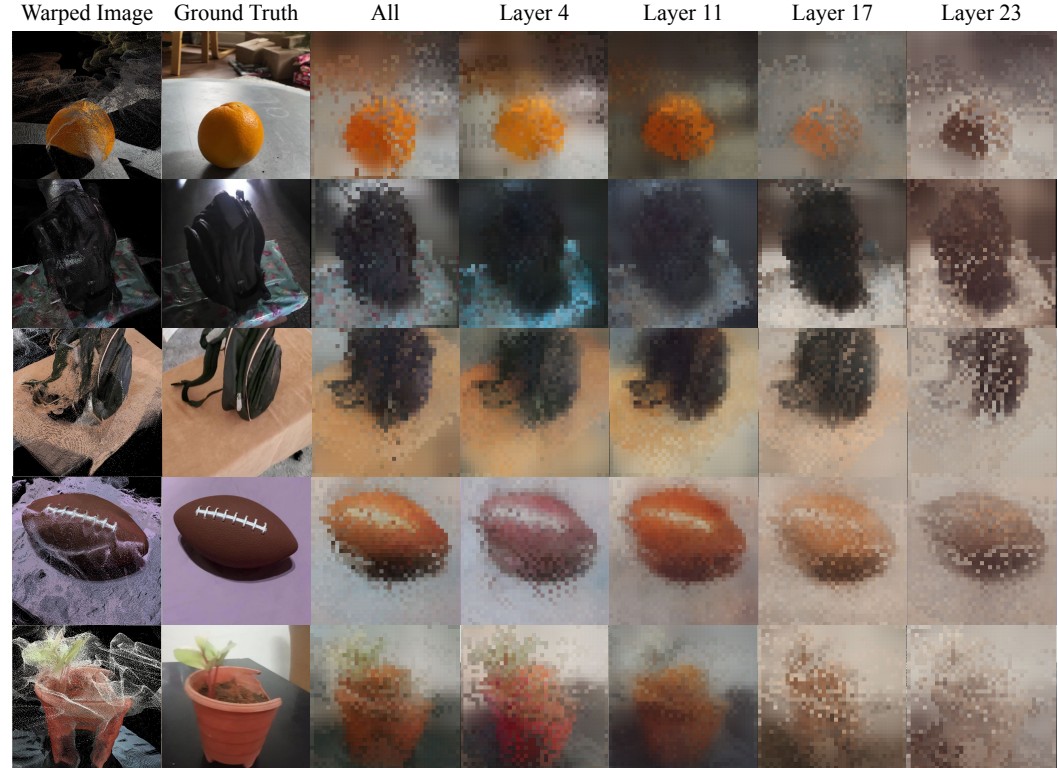

| Warped Image | Ground Truth | All | Layer 4 | Layer 11 | Layer 17 | Layer 23 |

Figure 12: **Per-layer probing qualitative results.** We visualize generation results using VGGT features extracted from individual layers and their combination. Early layers (4th, 11th) retain rich semantic information, producing semantically coherent images with accurate color and texture. In contrast, deeper layers (17th, 23rd) emphasize geometric structure but lack semantic detail. Combining features across all layers yields the most faithful reconstructions, achieving both structurally accurate and semantically realistic outputs.

## D   ADDITIONAL RESULTS

### D.1   ADDITIONAL COMPARISON

We compare our method against warping-and-inpainting approaches using single reference image, specifically LucidDreamer (Chung et al., 2023) and GenWarp Seo et al. (2024). Evaluation is conducted on the DTU dataset (Jensen et al., 2014), which was excluded from training for all methods, thereby demonstrating zero-shot generalization capabilities. To ensure fair comparison, all warping-and-inpainting methods utilize VGGT Wang et al. (2025) as the shared

| Methods | PSNR↑ | SSIM↑ | LPIPS↓ |
|---|---|---|---|
| LucidDreamer Chung et al. (2023) | **12.96** | 0.248 | 0.385 |
| GenWarp Seo et al. (2024) | 8.69 | 0.253 | 0.597 |
| **ReNoV (Ours)** | 12.63 | **0.443** | **0.261** |

Table 5: **Comparison with other warping-and-inpainting models**. We compare our model against LucidDreamer Chung et al. (2023) (using SD-Inpainting Rombach et al. (2022)) and Gen-Warp Seo et al. (2024).

geometry prediction model. Quantitative results in Table 5 demonstrate that our framework achieves superior performance in SSIM and LPIPS, maintaining competitive results in PSNR.

### D.2   ABLATION STUDY

**Qualitative results.**   Fig. 13 shows additional ablation results for three configurations: (a) baseline with semantic-only conditioning; (b) baseline + explicit geometric guidance via pointmaps; (c) ours with implicit semantic and geometric conditioning using VGGT Wang et al. (2025) features. Conditioning on VGGT features enables the model to achieve more accurate reconstructions and plausible inpainting by leveraging rich implicit geometric and semantic information. In contrast,

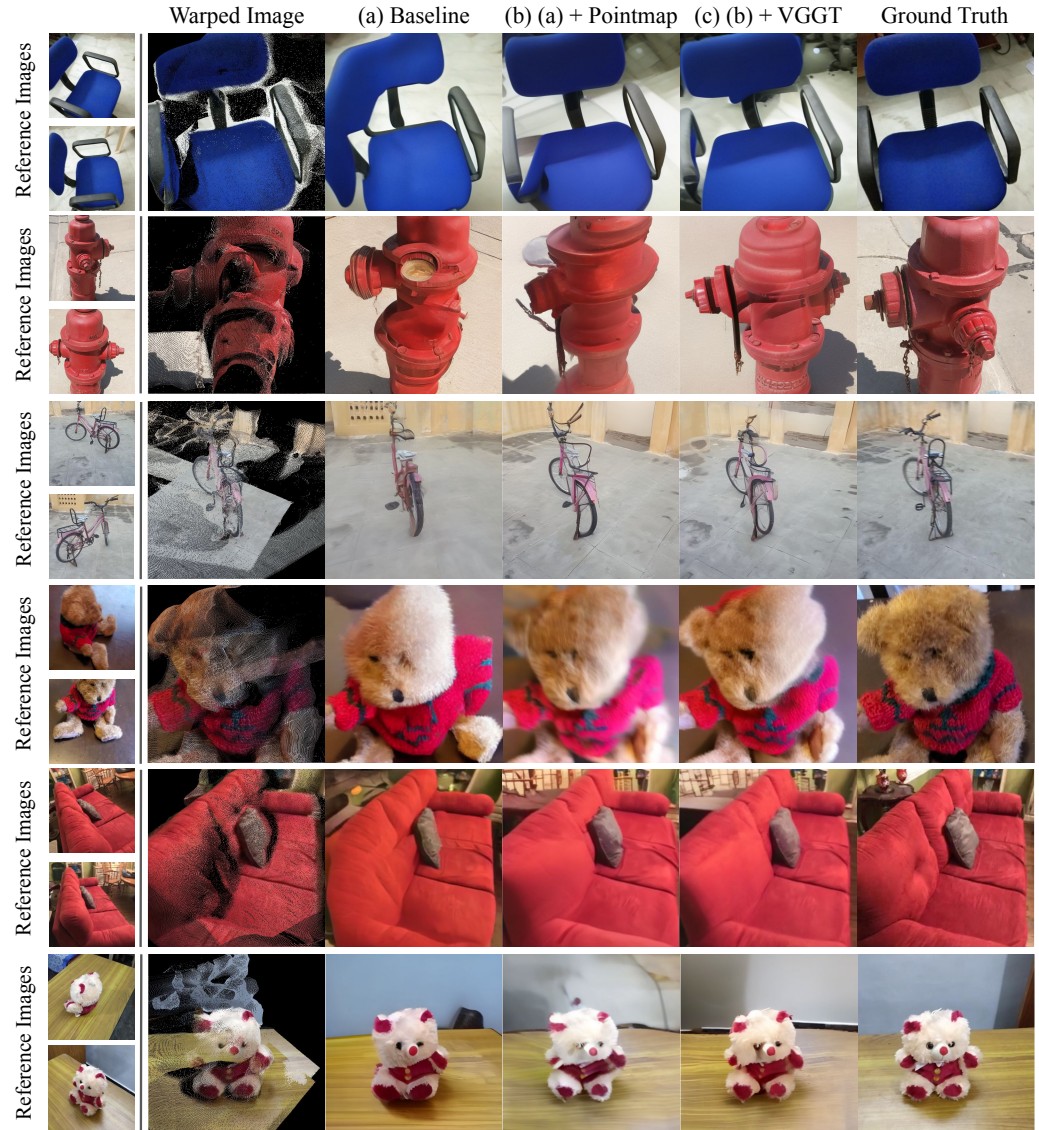

Figure 13: **Qualitative ablation results.** (a) *Baseline*: Lacks geometric guidance, resulting in misaligned structures (e.g., distorted chair, missing bicycle wheel, and incomplete teddy bear). (b) *Baseline + pointmap*: Improves geometric alignment but suffers from distortion due to noisy geometry and inaccurate inpainting (e.g., deformed chair seat). (c) *Ours with VGGT features*: Implicit semantic and geometric conditioning enables accurate reconstruction of visible regions and plausible inpainting of occluded areas.

(a) and (b) exhibit noticeable geometric distortions and incomplete inpainting, highlighting the limitations of lacking or noisy geometric cues.

**Attention map visualization.** We further analyze the cross-view attention maps of the denoising U-Net trained under configurations (a), (b), and (c). As shown in Fig. 14, the baseline model (a) attends to geometrically and semantically misaligned regions in the reference images, leading to inaccurate reconstruction and inpainting. Explicit geometric guidance via pointmaps (b) partially reduces this misalignment but remains insufficient due to noisy and incomplete geometric correspondences. In contrast, our final model conditioned on VGGT Wang et al. (2025) features (c) accurately attends to geometrically and semantically consistent regions in the reference views, significantly enhancing the quality of synthesized images. This confirms that VGGT features effectively guide cross-view

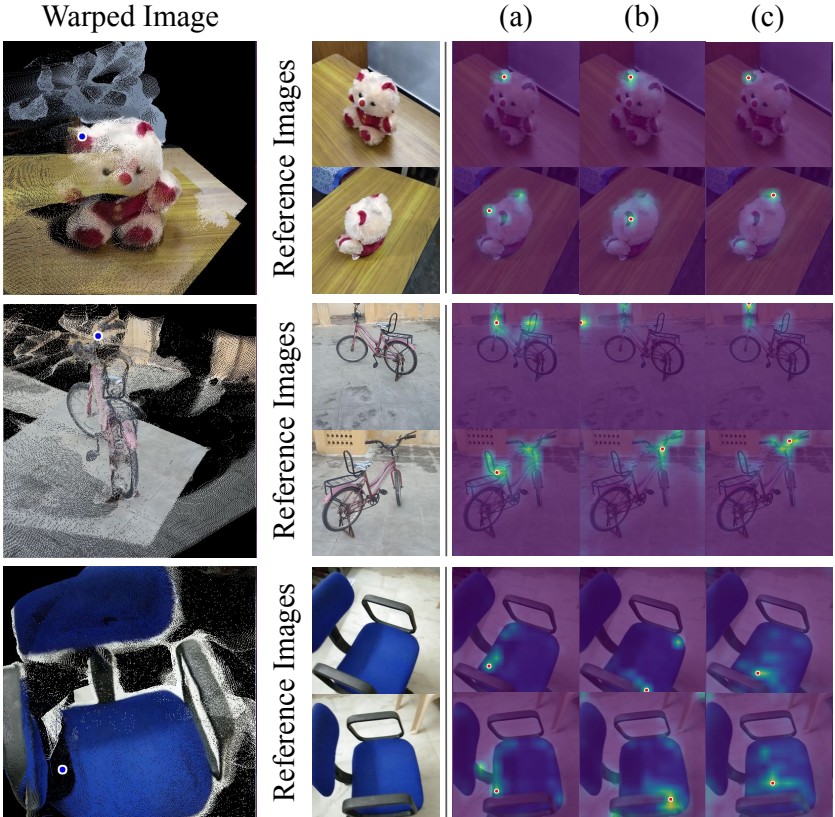

Figure 14: **Attention map visualization for ablation study.** The leftmost column shows a query point (blue dot) in the warped image, with corresponding cross-attention maps over reference images shown on the right. Configurations (a) and (b) attend to incorrect regions for both reconstruction (e.g., teddy bear's ear, bicycle handle) and inpainting (e.g., chair seat), due to limited or noisy geometric guidance. In contrast, VGGT-based conditioning (c) guides attention to geometrically and semantically aligned regions, accurately distinguishing fine structures such as the correct ear of the teddy bear.

attention toward optimal reference positions by implicitly encoding comprehensive geometric and semantic correspondences.

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
