# OpenReview forum: "Feature Warping-and-Conditioning for Representation-Guided Novel View Synthesis"
_ICLR.cc/2026/Conference — ICLR 2026 Conference Withdrawn Submission_

### Official Review · Reviewer_7QwP · 2025-10-30

**Soundness:** 3
**Presentation:** 3
**Contribution:** 2
**Rating:** 6
**Confidence:** 3

**Summary:**

a diffusion-based novel view synthesis method that leverages VGGT features through feature warping-and-conditioning. it achieves superior performance on RealEstate10K and DTU, with ablations confirming VGGT's critical role in maintaining geometric and semantic consistency.

**Strengths:**

- the idea of using VGGT features for conditioning in a diffusion-based warping-and-inpainting framework is novel and well-motivated.
- the experimental setup is rigorous, with strong baselines, multiple datasets, and both interpolation and extrapolation settings.
- well-organized, with clear explanations and illustrative visualizations.
- the method advances the state of the art in novel view synthesis, particularly for extrapolative scenarios where generative capabilities are crucial.

**Weaknesses:**

- does not discuss computational cost or inference speed, which are important for practical deployment.
- while VGGT features are shown to be effective, the paper does not explore whether similar results could be achieved with other multi-view consistent features or through fine-tuning existing models.
- the comparison with other warping-and-inpainting methods is limited and could be expanded.

**Questions:**

- how does ReNoV scale with the number of reference views? is there a diminishing return or a computational bottleneck?
- could the proposed feature conditioning mechanism be applied to other generative tasks beyond novel view synthesis?
- were any alternative multi-view feature extractors besides VGGT considered? if so, how did they compare?
- the method relies on an off-the-shelf geometry predictor. how sensitive is the performance to the quality of the estimated pointmaps and poses?
- have you considered evaluating on dynamic scenes or video sequences, and if so, what challenges arise?

---

### Official Review · Reviewer_S48k · 2025-10-31

**Soundness:** 2
**Presentation:** 3
**Contribution:** 2
**Rating:** 2
**Confidence:** 5

**Summary:**

The paper introduces ReNoV, a diffusion-based framework for novel view synthesis that reformulates the task as feature warping and inpainting. By leveraging VGGT for geometry-aware and semantically rich conditioning, the model synthesizes novel views without explicit 3D reconstruction or known camera poses.

**Strengths:**

Clear and elegant formulation connecting geometry and diffusion.

Innovative use of VGGT features for semantic–geometric conditioning.

**Weaknesses:**

Cannot handle dynamic scenes and performance is bounded by VGGT.

Missing many SoTA NVS methods for comparison, e.g. SEVA, ViewCrafter, CausNVS, CAT3D, LVSM.

The paper shows 1-3 views as input, how many input and ouput views can the model handle? How does the computation cost scale with input/output view number?

Missing the training details, e.g. how many GPU (hours)? What are the training settings, e.g., batch size, model size?

**Questions:**

Please see the weakness.

---

### Official Review · Reviewer_4WZN · 2025-10-31

**Soundness:** 2
**Presentation:** 2
**Contribution:** 2
**Rating:** 4
**Confidence:** 4

**Summary:**

The paper presents a diffusion-based novel-view synthesis framework that utilizes VGGT's multi-view geometry features for improved image reconstruction and inpainting. It conducts extensive ablation studies to demonstrate the effectiveness of implicit semantic and geometric conditioning, showing that the proposed method outperforms baseline models in terms of structural coherence and semantic fidelity.

**Strengths:**

1. The integration of VGGT features enhances the model's performance in novel-view synthesis, as evidenced by quantitative results.
2. Ablation studies provide clear insights into the contributions of different components, validating the proposed approach.

**Weaknesses:**

1.	The proposed ReNoV framework relies heavily on existing warping-and-inpainting paradigms (e.g., GenWarp, Seo et al., 2024) and ControlNet-style dual U-Net architectures. The core innovation—integrating VGGT features into diffusion-based view synthesis—lacks sufficient technical distinctiveness:
2.	The comparison with "non-generative models" (PixelSplat, MVSplat) in Tables 2–3 is misleading. Non-generative methods (e.g., MVSplat) are inherently limited in inpainting occluded regions, so outperforming them does not demonstrate ReNoV’s strengths relative to generative peers.
3.	Computational Efficiency: No analysis of inference time or memory usage is provided. Given that VGGT (a transformer-based geometry model) and Stable Diffusion 2.1 are both computationally heavy, ReNoV’s practicality for low-resource settings (e.g., edge devices) is unclear. The paper also does not compare efficiency to lightweight methods like NopoSplat.
4.	This paper discusses the sparse view novel view synthesis. Is it able to synthesize novel views based on a single view, because in real-world scenarios, we usually can collect a single view image[1][2][3].

[1] Liu, Ruoshi, et al. "Zero-1-to-3: Zero-shot one image to 3d object." Proceedings of the IEEE/CVF international conference on computer vision. 2023.
[2] Yang, Y., Qiu, Z., Zhang, S., & Tan, M. Disparity Guidance and Spatial-Angular Interaction for Single-View-Based Light Field Synthesis. Available at SSRN 5034806.
[3] Jiang, Lei, Gerald Schaefer, and Qinggang Meng. "Multi-scale feature fusion for single image novel view synthesis." Neurocomputing 599 (2024): 128081.

**Questions:**

Please refer to the weakness part.

---

### Official Review · Reviewer_QMVB · 2025-11-01

**Soundness:** 2
**Presentation:** 3
**Contribution:** 2
**Rating:** 2
**Confidence:** 3

**Summary:**

This paper proposes Representation-guided Novel View synthesis, a framework for diffusion-based novel view synthesis that operates on sparse and unposed image collections. The core idea is to reformulate NVS as a warping-and-inpainting task, guided by the rich geometric and semantic features from VGGT.  It first uses VGGT to estimate camera poses and extract multi-scale features from a set of reference images. A diffusion model, architecturally similar to ControlNet, then generates the final image. It is conditioned on both the warped features and the original reference features.

**Strengths:**

1. Defining the task of scene-level reconstruction with sparse and unposed images as a warping-and-inpainting problem is intuitive. By leveraging existing methods such as VGGT to obtain a strong prior and then using diffusion models to generate the unknown regions, the approach is both reasonable and effective.
2. By modeling a sparse collection of images without pre-existing camera poses, the method becomes more flexible in its application scenarios, which is largely attributed to the capabilities of VGGT.
3. Both the visual and quantitative results show significant improvements over the baseline.

**Weaknesses:**

1. The method directly combines the VGGT model with a diffusion model, injecting multi-view conditions into the diffusion model using a ControlNet-like architecture. This approach is already widely adopted and does not present any particularly creative or novel design.
2. Heavy Dependency on VGGT: The entire framework's performance is fundamentally bottlenecked by the quality of the VGGT model. Any errors in VGGT's pose estimation or feature extraction will directly propagate and negatively impact the final generated image.

**Questions:**

Nil

---

### Note · Authors · 2026-01-05

I have read and agree with the venue's withdrawal policy on behalf of myself and my co-authors.